# Genomic and transcriptomic alterations associated with drug vulnerabilities and prognosis in adenocarcinoma at the gastroesophageal junction

Yuan Lin[1,8], Yingying Luo [2,8], Yanxia Sun [2,8], Wenjia Guo [2,3,8], Xuan Zhao[2,8], Yiyi Xi[2], Yuling Ma [2], Mingming Shao[2], Wen Tan[2], Ge Gao [1,4✉], Chen Wu [2,5,6✉] & Dongxin Lin[2,5,7]

Adenocarcinoma at the gastroesophageal junction (ACGEJ) has dismal clinical outcomes, and there are currently few specific effective therapies because of limited knowledge on its genomic and transcriptomic alterations. The present study investigates genomic and transcriptomic changes in ACGEJ from Chinese patients and analyzes their drug vulnerabilities and associations with the survival time. Here we show that the major genomic changes of Chinese ACGEJ patients are chromosome instability promoted tumorigenic focal copy-number variations and COSMIC Signature 17-featured single nucleotide variations. We provide a comprehensive profile of genetic changes that are potentially vulnerable to existing therapeutic agents and identify Signature 17-correlated IFN-α response pathway as a prognostic marker that might have practical value for clinical prognosis of ACGEJ. These findings further our understanding on the molecular biology of ACGEJ and may help develop more effective therapeutic strategies.

[1] Beijing Advanced Innovation Center for Genomics (ICG), Biomedical Pioneering Innovation Center (BIOPIC), Peking University, Beijing, China. [2] Department of Etiology and Carcinogenesis, National Cancer Center/National Clinical Research Center/Cancer Hospital, Chinese Academy of Medical Sciences and Peking Union Medical College, Beijing, China. [3] Cancer Institute, Affiliated Cancer Hospital of Xinjiang Medical University, Urumqi, China. [4] State Key Laboratory of Protein and Plant Gene Research, School of Life Sciences, Center for Bioinformatics, Peking University, Beijing, China. [5] Collaborative Innovation Center for Cancer Personalized Medicine, Nanjing Medical University, Nanjing, China. [6] CAMS Key Laboratory of Genetics and Genomic Biology, Chinese Academy of Medical Sciences and Peking Union Medical College, Beijing, China. [7] Sun Yat-sen University Cancer Center, State Key Laboratory of Oncology in South China, Guangzhou, China. [8] These authors contributed equally: Yuan Lin, Yingying Luo, Yanxia Sun, Wenjia Guo, Xuan Zhao. ✉email: gaog@mail.cbi.pku.edu.cn; chenwu@cicams.ac.cn

In the past few decades, the incidences of adenocarcinoma at the gastroesophageal junction (ACGEJ) are rapidly increasing worldwide[1,2]. While it can be categorized as esophageal or gastric adenocarcinoma[3], ACGEJ has been treated as the latter in Chinese hospitals because esophageal cancer in China is mostly squamous-cell carcinomas[4]. Surgical resection is the standard treatment for early-stage ACGEJ and for locally advanced or unresectable tumors, pre- and post-operative chemotherapies are often used[5]. Targeted therapies are only for patients with late-stage metastatic HER2-positive tumors[6], although Nivolumab, an immuno-oncology agent targeting PD-L1, has showed encouraging efficacy in patients with unresectable advanced or recurrent HER2-negative ACGEJ[7]. The 5-year survival rates of this cancer are 20−25%, lower than that of esophageal or gastric cancers. Thus, it is urgent to apprehend molecular characteristics that can serve as potential drug targets and (or) prognostic indicators.

In a previous study conducted in Caucasian patients by the Cancer Genome Atlas (TCGA) project[8], 65% of ACGEJ were categorized as the chromosomal instability (CIN) subtype of gastric cancer, characterized by preponderating focal copy-number variations (CNVs) in the tumor genome. Somatic single nucleotide variations (SNVs) and small insertions and deletions (indels) significantly recur in *TP53* and genome-wide single base substitutions (SBSs) typically form a pattern known as Signature 17 in the Catalogue of Somatic Mutations in Cancer (COSMIC)[9,10]. Signature 17 may be the footprint of an early mutational mechanism initiating esophageal and gastric adenocarcinoma[11], and its characteristic SBS, 5′-C[T > G]T-3′, has been associated with poor survival of esophageal adenocarcinoma[12].

Due to different genetic makeup and environments including lifestyles, whether ACGEJ genomes in Chinese patients share the above characteristics is unclear. Previous gastric cancer studies on East Asian patients have each collected about 30 ACGEJ samples[13,14], probably too few to draw rigorous conclusions. Moreover, previous studies did not provide enough information on transcriptomic changes that could further their significant findings on the genomic changes of ACGEJ. In the present study, we have assembled a relatively large set of Chinese ACGEJ patients and have sequenced genomes and transcriptomes of matched tumor and adjacent normal tissue samples. By jointly analyzing these data and comparing with the findings in Caucasian patients, we show that ACGEJ in Chinese patients are also dominated by CIN-associated focal CNVs and that Signature 17 activities correlate with multiple essential genomic and transcriptomic changes of ACGEJ. Furthermore, we deliver a comprehensive profile of genetic alterations potentially vulnerable to existing treatments, and identify genomic and transcriptomic prognostic markers of potential clinical values. These findings may improve current knowledge about ACGEJ and contribute to its precision diagnosis and treatment.

## Results

**CIN-associated focal CNV is the major feature of ACGEJ genomic alterations.** We performed whole-genome sequencing (WGS) on ACGEJ tumor samples containing ≥60% of cancer cells (Supplementary Fig. 1a) and matched blood samples from 124 Chinese patients (mean coverage 61x and 31x, respectively) and identified 2,558,269 SNVs and 1,258,899 indels. The tumor mutation burden (TMB) ranged from 0 to 13.1 (median 1.8) per megabase (Mb). The most significantly mutated gene was *TP53* (FDR $q \le 0.05$) with coding mutations found in 71.0% (88/124) samples. To increase the detection power, we combined the reported coding-region mutations data of 151 ACGEJ samples[8,15,16] with our data and found that *TP53* was the only gene recurrently mutated in ≥10% of samples (Supplementary

Table 1). CNVs overlapped with a median of 7.8% of the tumor genome and 8.4% of the coding region, respectively (Supplementary Fig. 2a) and the jagged layout of genome-wide CNV distributions (Fig. 1a) indicates widespread focal CNVs. Protein-coding genes subject to CNVs were 5.7 times (median) more than those altered by non-silent SNVs/indels (Fig. 1b). These results suggest that focal CNVs are the major genomic alterations of ACGEJ in Chinese patients.

Since 65% of TCGA ACGEJ samples were deemed CIN and CIN is a common source of CNVs in cancer genomes associated with metastasis, therapeutic resistance, and immune evasion[17,18], we investigated the prevalence of CIN in our samples. We found a published CIN gene signature (CIN70)[19] significantly over-expressed in tumors compared with adjacent normal tissues ($P = 0.001$; Fig. 1c) and whole-genome doubling (WGD), a known precursor to CIN[20], in 59.7% (74/124) of our ACGEJ genomes. We also found other genomic abnormalities suggestive of CIN including chromothripsis ($n = 77$, 54 with WGD), kataegis ($n = 74$), and complex structural variations (SVs) such as translocations (median 141.5 per ACGEJ genome) and inversions (median 257.5 per tumor genome) (Supplementary Data 1, 2). The expression levels of CIN70 were significantly higher in WGD than in non-WGD tumor genomes (median 0.50 versus 0.28, $P = 2.15e-5$) and significantly correlated with the number of chromosomal arm and gene level CNVs (Spearman's $\rho = 0.30$ and 0.27, $P = 6.60e-4$ and 0.003, respectively) (Fig. 1d). ACGEJ genomes with WGD also had more gene level CNVs than those without WGD (median 609.5 versus 312, $P = 8.01e-4$). The median ploidy of WGD genomes was 3.1 and they had a significantly larger proportion of autosomal genome losing heterozygosity than non-WGD genomes (39% versus 25%, $P = 2.43e-6$), indicating frequent single-copy losses after WGD. Together, these results suggest the abundant CNVs observed in our ACGEJ genomes were associated with CIN.

The genomic regions of significantly recurrent CNVs (FDR $q \le 0.1$, in ≥10% samples) harbored 25 oncogenes or tumor suppressor genes (TSGs) knowingly affected by CNVs[21,22] (Fig. 1e) including *CCNE1*, *RICTOR*, *VEGFA*, *ERBB2*, *FGFR2*, *BCL2L1*, *CDK6*, *ERBB3*, *MET*, *CDH1*, *ARID1A*, *APC*, and *CDKN2B* that had correlated copy-number and expression changes (Spearman's $\rho \ge 0.3$, FDR $q < 0.002$) (Supplementary Fig. 2b). CNVs of these 13 genes are potential ACGEJ drivers, of which *CCNE1* and *ERBB2* amplifications occurred most frequently ($n = 35$ and 24; 28.2% and 19.4%, respectively) and significantly co-existed ($n = 13$, 10.5%; $P = 0.004$). We found genomic and transcriptomic evidence suggesting an association between the dysfunction of *CCNE1* and CIN. ACGEJ with *CCNE1* copy number gains ($n = 67$, 54.0%) had more CNVs at both chromosomal arm level (median 21 versus 14, $P = 3.20e-5$) and gene level (median 618 versus 251, $P = 8.88e-5$), and were more likely to undergo WGD ($P = 4.28e-4$) than ACGEJ without *CCNE1* gains ($n = 57$, 46.0%) (Fig. 1f); these associations remained significant after adjusting for *TP53* mutation status ($P < 0.02$). The expression levels of *CCNE1* were highly correlated with CIN70 activities (Spearman's $\rho = 0.53$, $P = 3.93e-10$) and significantly elevated in WGD tumor samples (fold change = 1.21, $P = 0.011$) (Fig. 1g). Consistently, *CCNE1* amplification has been associated with WGD in TCGA pan-cancer analyses[23,24] and *CCNE1* overexpression has been shown to induce CIN phenotypes in various cancer cells[25,26].

**COSMIC Signature 17 is the characteristic ACGEJ mutational signature.** We next examined the mutation spectra of our ACGEJ genomes to characterize the mutational signature and found 5′-C[T > G]T-3′ was the most common somatic SBS across the

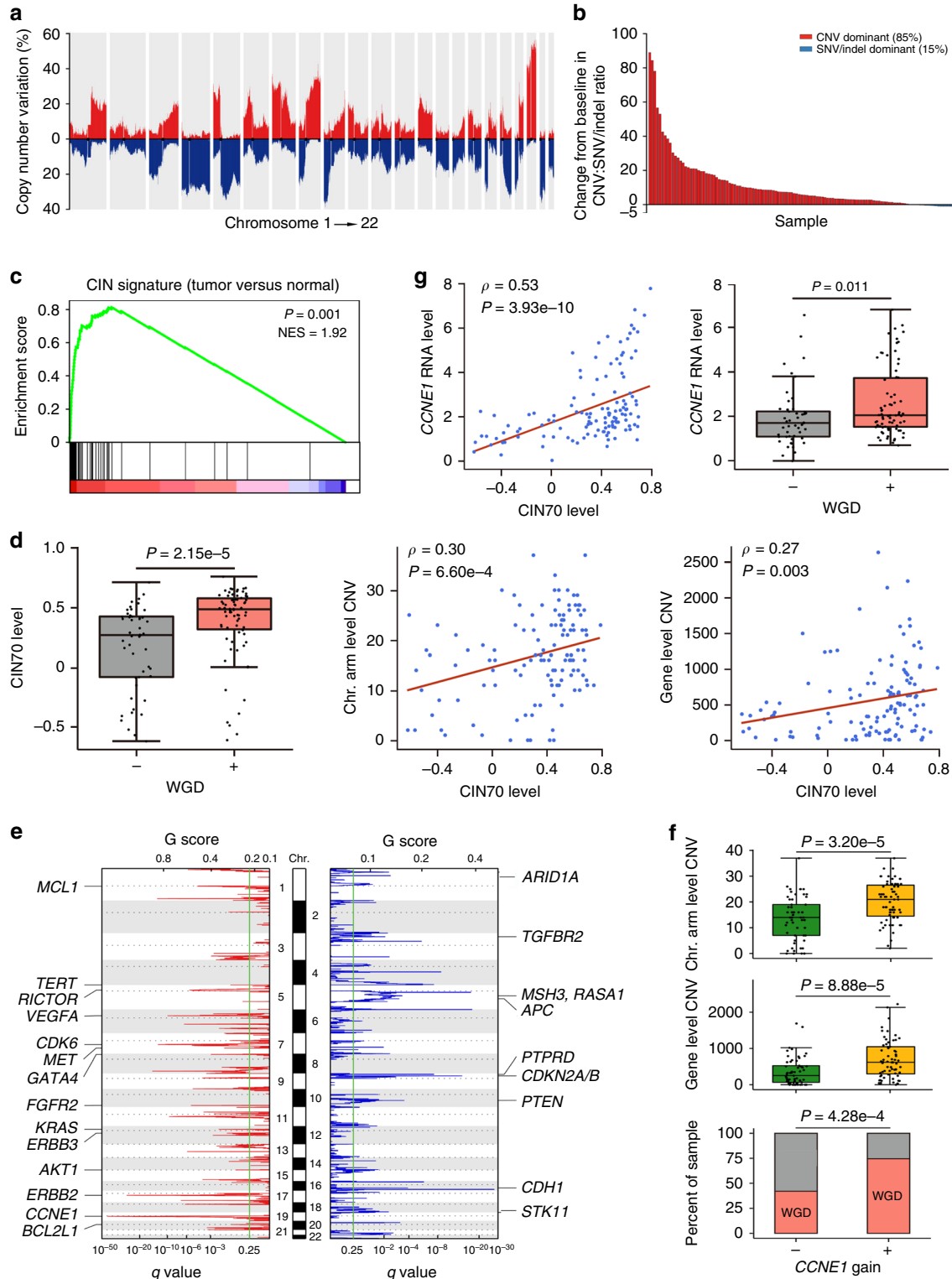

genomes (Fig. 2a). When fitted to reported COSMIC mutational signatures, this SBS becomes a major component of Signature 17. We also found recurrent activities of COSMIC Signatures 1, 3, 5, and 8 (Fig. 2b). In the ACGEJ genomes of our patients, 52.3% (257,673/493,106) of Signature 17 attributed SNVs were located at intergenic regions. We searched for potential cancer-driving regulatory elements in these regions and found Signature 17 SNVs in 75.3% (314/417) of significantly mutated *CTCF* binding sites[27] (Supplementary Data 3). Because oncogenic SNVs at

*CTCF* binding sites have been linked to CIN[27–29], we then investigated correlations between Signature 17 and CIN-related genomic alterations. Signature 17 activities were higher in ACGEJ genomes with *TP53* coding mutations or WGD or chromothripsis than in genomes without these features ($P = 0.002$, 0.006, and 0.009, respectively) and were positively correlated with TMB and the number of chromosomal arm level CNVs (Spearman's $\rho = 0.26$ and 0.32, $P = 0.004$ and 2.57e-4, respectively; Fig. 2c, d).

**Fig. 1 Featured ACGEJ genomic and transcriptomic changes. a** Frequencies of CNVs detected in 124 ACGEJ samples, with gains in red and losses in blue. Bars represent non-overlapped 1-million-base-pair windows along the genome. **b** Bar plot comparing the number of genes altered by CNVs and SNVs/indels in each of 120 ACGEJ genomes with both types of alterations. The $y$-axis indicates the ratio of CNVs to SNVs/indels minus 1. Only deletions, amplifications, or non-silent coding mutations are counted. Most samples (102/120, 85%) show a preponderance of CNVs over SNVs/indels. **c** GSEA analysis comparing CIN70 activities in ACGEJ samples and adjacent normal tissue samples. The plot shows overexpressed CIN70 in ACGEJ samples, the normalized enrichment score (NES) and the $P$ value. **d** Associations between CIN70 activities and the WGD status ($+$ and $-$ indicating 74 and 50 tumor genomes with and without WGD, respectively; two-sided Wilcoxon rank-sum test), the number of chromosomal (Chr.) arm or gene level CNVs (Spearman's correlation tests) in ACGEJ samples. **e** GISTIC2.0 identified recurrent focal CNVs in 124 ACGEJ genomes, with 25 potential CNV drivers annotated on the plot. **f** Box and bar plots comparing chromosomal (Chr.) arm level CNVs, gene level CNVs, and the frequencies of WGD in tumor genomes with and without *CCNE1* copy number gains ($n = 67$ and 57, respectively; two-sided Wilcoxon rank-sum tests). **g** Associations between *CCNE1* expression levels and CIN70 activities (Spearman's correlation test) or WGD status (two-sided Wilcoxon rank-sum test) in ACGEJ samples. Box plots in (**d, f, g**) show the median (central line), the 25–75% interquartile range (IQR) (box limits), the ±1.5 times IQR (Tukey whiskers), and all data points, among which the lowest and the highest points indicate minimal and maximal values, respectively.

We further investigated Signature 17-correlated transcriptomic changes. Since the underlying mutational process represented by Signature 17 has not been well established, we sought for genes whose expressions are correlated with Signature 17 activities and assessed the enrichment of these genes in cancer hallmark pathways and gene signatures (see Methods). Among pathways and gene signatures differentially expressed between our tumor and adjacent normal samples (FDR $q \leq$ 1e-4), the CIN70 gene signature was enriched with genes positively correlated with Signature 17 activities (e.g., *TPX2*, *NEK2*, and *KIF4A*). Additionally, the mitotic cell-cycle, mitotic spindle, E2F targets, and G2M checkpoint pathways were also enriched with genes positively correlated with Signature 17 (e.g., *MYBL2*, *KIF15*, and *CCNE1*), suggesting elevated cell proliferation of tumors with high Signature 17 activities. Immunity-related pathways including allograft rejection, interferon (IFN)-α or IFN-γ response, and the IL6-JAK-STAT3 signaling were down-regulated in tumor samples and they were enriched with Signature 17 negatively correlated genes (e.g., *CXCR3*, *A2M*, and *FGL2*), suggesting suppressed immune response in Signature 17-high tumors. We also found the oxidative phosphorylation and glycolysis pathways were, respectively, enriched with Signature 17 negatively and positively correlated genes (e.g., *CLDN3*, *NDUFA4L2*, and *UQCRC2*), suggesting metabolic remodeling in Signature 17-high tumors (Fig. 2e).

Similarly, assessing the enrichment of Signature 17-correlated genes in a set of marker genes for CD8 + T, γδ T, and natural killer (NK) cells[30] revealed decreased cytotoxic cell activities in Signature 17-high tumors (Fig. 2e). We estimated the fractions of ten immune cell types to total cells in ACGEJ and adjacent normal samples using our bulk RNA sequencing data and found CD8 + T and NK cells present (>0%) in 97.6% (120/123) and 98.4% (121/123) tumor samples, respectively, but the most prevalent immune cell types in tumors were immune suppressive regulatory T cells (Treg, median 5.49%). Signature 17 activities were negatively correlated with the faction of CD8 + T cells (partial Spearman's $\rho = -0.16$, $P = 9.95$e-6). Compared with normal samples, tumor samples had significantly increased fractions of CD8 + T (1.46–1.80%, $P = 0.006$) and NK cells (0.80–1.37%, $P = 3.65$e-13), and decreased fractions of neutrophils (4.28–3.07%, $P = 5.44$e-6) (Fig. 2f). Although the proportions of M1 and M2 macrophages both increased (1.52–4.76% and 1.60–2.79%, $P = 1.96$e-20 and 4.04e-5, respectively), the M1/M2 ratio was significantly higher in tumor than in normal samples (1.70 versus 0.71, $P = 1.15$e-13). However, immune checkpoint genes including *IDO1*, *HAVCR2*, *PDCD1LG2*, *CD274*, *CTLA4*, and *TIGIT* were significantly up-regulated in tumor samples (fold change >1.30, FDR $q \leq 1.0$e-5). These results collectively point to an immunosuppressive microenvironment of ACGEJ, which is even worse in Signature 17-high tumors.

**Comparative analysis on major genomic alterations of our patients and TCGA patients.** We compared clinically relevant characteristics of ACGEJ samples collected by this study ($n = 124$) and by TCGA ($n = 105$). Due to the controlled access of TCGA Level 1/2 data, we downloaded Level 3 data and applied the same downstream analyses. The patients enrolled in this study were slightly younger than TCGA patients (median 65 and 66 years, respectively, $P = 0.050$) and exhibited more advanced (stage III or IV) ACGEJ (72 and 41 patients, respectively, $P = 0.021$). While two cohorts shared similar sex distributions, the TCGA cohort was more ethnically diverse. Eighty-eight ancestry-known TCGA patients contain 78 Caucasians, nine Asians, and one African American. Five TCGA ACGEJ samples were likely microsatellite instable, while no such samples were identified in our cohort. Disregarding these samples, the TMB of the rest of TCGA ACGEJ ($n = 100$) was still higher than that of ours (median 2.68 and 1.84, respectively). The recurrence rates of functional mutations (annotated as oncogenic or likely oncogenic by OncoKB[21]) in *TP53*, *PTEN*, *ARID1A*, *CDKN2A*, and *KRAS* appeared to be significantly higher in TCGA patient cohort than in our patient cohort. We did not find *CDKN2A* and *KRAS* mutations in our samples but detected 4.0% of *LIPF* mutations absent in TCGA samples (Supplementary Table 2).

Our samples were not significantly different from TCGA samples in the number of chromosomal arm level CNVs (median 17 for both) or gene level CNVs (median 423.5 and 370, respectively). High correlations of the cohort-level recurrences of arm level CNVs (Fig. 3a) indicated similar arm level CNV patterns across two cohorts. The genome-wide distributions of focal CNV regions were also similar in terms of recurrences and amplitudes (Fig. 3b). We assessed the overlaps between significantly recurrent focal CNVs (FDR $q < 0.25$) across two cohorts and found 46.4% of focal amplifications and 50.8% of focal deletions identified from our samples had ≥50% overlap with those identified from TCGA samples. We then searched for CNV driver genes in highly recurrent focal CNV regions identified in either cohort; any two regions identified from different cohorts and reciprocally overlapping at ≥50% of both their sizes were merged into one big region and considered shared by both cohorts. In these regions, we found 28 genes with highly correlated copy-number and expression changes (Spearman's $\rho \geq 0.3$, FDR $q < 0.05$) in respective cohorts (Fig. 3c). Homologous deletions or amplifications of these genes, which might drive ACGEJ, occurred mostly at comparable frequencies between two cohorts (Fig. 3d), except for more frequent *CCNE1* and *BCL2L1* amplifications in our samples ($P = 0.016$ and $P = 0.007$, respectively).

We then compared the impacts of potentially functional alterations (i.e., oncogene amplification, TSG homologous deletion, and functional mutations in both) at the pathway level. In

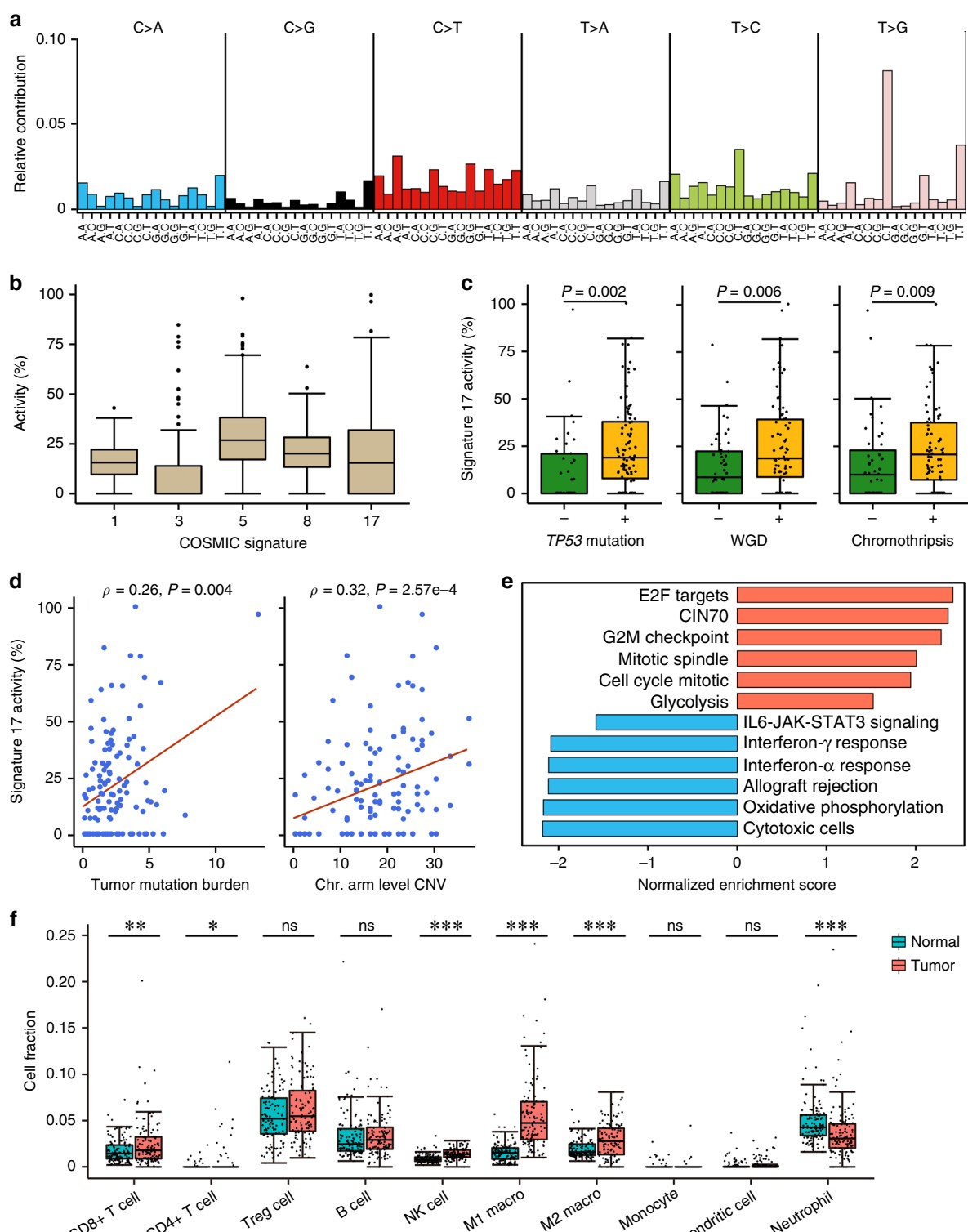

**Fig. 2 Genomic and transcriptomic features associated with COSMIC Signature 17. a** Mutational spectra of our ACGEJ samples. Different colors represent six SBS categories. On the *x*-axis, 16 possible trinucleotide contexts are repeated for each SBS category. **b** Distributions of COSMIC Signatures 1, 3, 5, 8, and 17 activities in 124 ACGEJ genomes. **c** Box plots comparing Signature 17 activities in ACGEJ samples with and without *TP53* mutations (*n* = 88 and 36, respectively), WGD (*n* = 74 and 50, respectively), or chromothripsis (*n* = 77 and 47, respectively); *P* values derived from two-sided Wilcoxon rank-sum tests. **d** Spearman's correlations between Signature 17 activities and TMB or chromosomal (Chr.) arm level CNVs in ACGEJ. **e** Pathways or gene signatures enriched with genes positively (red bars) or negatively (blue bars) correlated with Signature 17 activities. **f** Box plots comparing the fractions of ten immune cell types in total cells between 124 pairs of ACGEJ and matched adjacent normal tissue samples. *\*P* < 0.05; \*\**P* < 0.01; \*\*\**P* < 0.001; ns, not significant of two-sided Wilcoxon rank-sum test; macro., macrophage. Box plots show the median (central line), the 25–75% interquartile range (IQR) (box limits), the ±1.5 times IQR (Tukey whiskers), and outliers (**b**) or all data points (**c, f**); the lowest and the highest data points indicate minimal and maximal values, respectively.

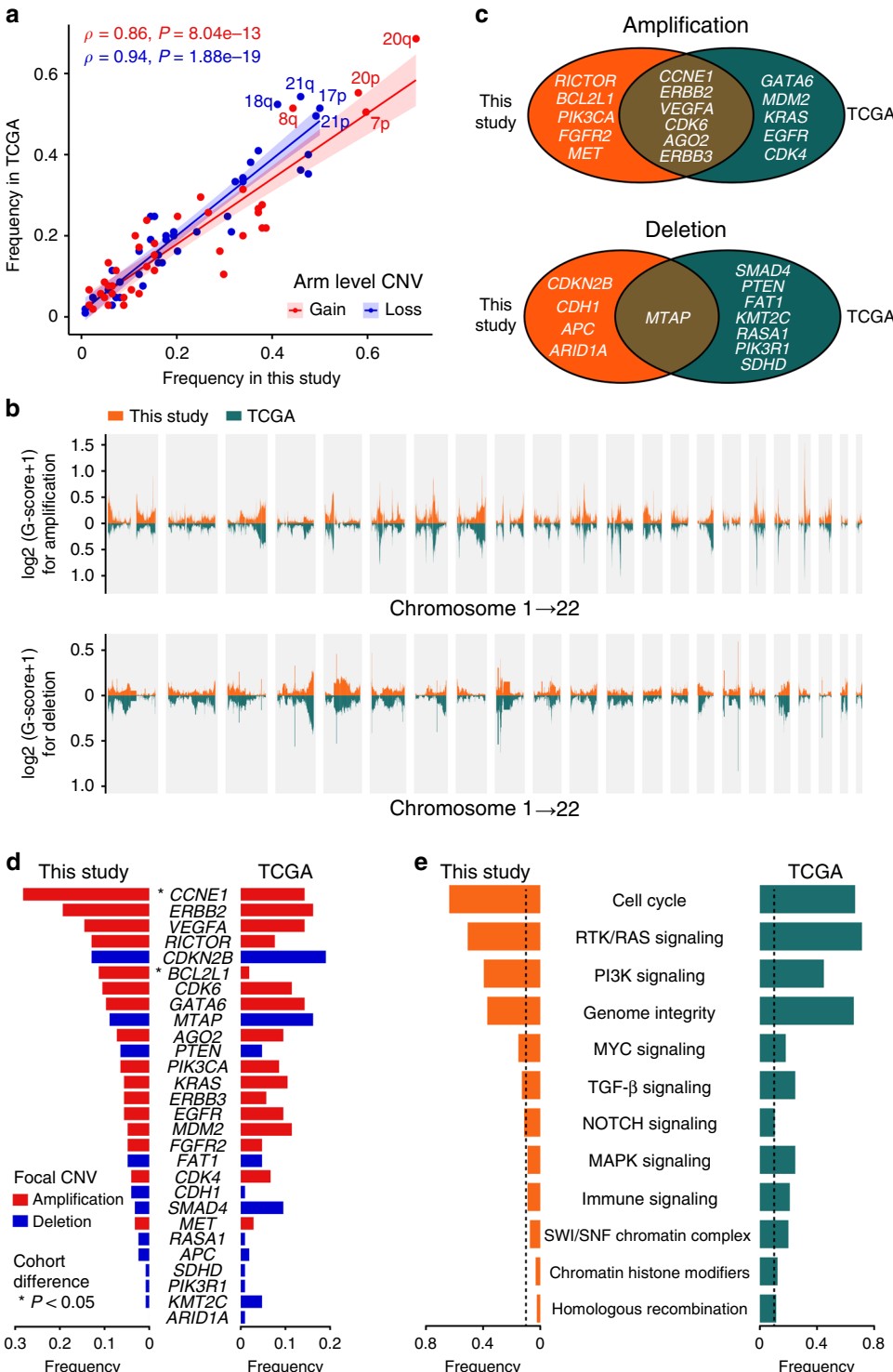

**Fig. 3 Comparison on genomic alterations between our patients and TCGA patients. a** Correlations of arm level CNV frequencies between our tumor samples and TCGA samples. Red and blue represent arm level gain and loss, respectively. **b** Bar plots comparing log2-tranformed GISTIC G-scores, a measure for both frequency and amplitude, of focal copy-number gain (upper panel) and loss (lower panel) regions in our tumor samples and TCGA samples. **c** Shared and unique putative CNV driver genes targeted by recurrent focal copy-number gains (upper panel) and losses (lower panel) in our samples and TCGA samples. **d** Bar plots comparing amplification (red) and deletion (blue) frequencies of putative CNV driver genes (**c**) across two cohorts. Genes with significantly different alteration frequencies in two cohorts (*P* < 0.05) were asterisked. **e** Bar plots comparing the frequencies of featured pathway perturbations across two cohorts. Dashed lines indicate 10% alteration frequency.

both cohorts, we found recurrent perturbations (>10% samples) to the cell-cycle, receptor tyrosine kinases (RTK)/RAS, PI3K, MYC, TGF-β, and NOTCH signaling pathways, and to genome integrity related genes (e.g., *TP53, MDM2,* and *ERCC2*), with

different rates (Fig. 3e). In addition, MAPK signaling (e.g., *KRAS, MAP2K1*), immune signaling (e.g., *HLA-A/B*), the SWI/SNF chromatin remodeling complex (e.g., *ARID1A, SMARCA4*), chromatin histone modifiers (e.g., *EP300, KMT2C*), and the

homologous recombination pathway (e.g., *BRCA1*, *TP53BP1*) were recurrently perturbed in the TCGA cohort but not in ours. Despite different genomic perturbation rates, the activities of these pathways or gene sets in our tumor samples and TCGA samples, as quantified by Gene Set Variation Analysis (GSVA)[31] scores using normalized and batch corrected gene expression data, were similar with no statistically significant difference. So were the estimated fractions of different immune cell types. Together, these results suggest that our patient set and TCGA patient set had similar genomic alterations with some differences.

**Genomic alterations indicate potential vulnerability to existing therapeutic agents.** The lack of specific and effective drugs for ACGEJ treatments is largely due to our limited knowledge on the genomic characteristics of this disease. Thus, we screened our ACGEJ samples for genomic alterations predicting vulnerabilities to therapeutic agents approved by FDA or NCCN, currently in clinical trials, or supported by evidence from pre-clinical assays or case reports[32]. This analysis revealed 488 alterations in 67 genes with potential responses to 66 classes of 93 drugs (Supplementary Data 4). We predicted the genetic vulnerabilities of 105 TCGA ACGEJ samples with the same approach (Supplementary Data 5). For cytotoxic chemotherapeutic agents, we predicted that 96 of our patients would respond to six classes of drugs, including Anthracyclines, Gemcitabine, Mytomycin C, Temozolomide, Platinum agents, and Vinblastine (Fig. 4a). Anthracyclines, Gemcitabine, and Mytomycin C might have substantially higher response rates in our patients (71.0−73.4%, 88–91 of 124) than in TCGA ACGEJ patients (49.5−57.1%, 52−60 of 105) (Fig. 4a). The vulnerabilities of our patients to the three drugs were predicted mainly based on coding mutations in *TP53*. Anthracyclines has also been tested in a clinical trial as an inhibitor of *TOP2A*[33], which was amplified in 11.3% (14/124) of our patients. We found significantly more frequent CIN-related genomic alterations (i.e., WGD, chromothripsis, and kataegis) in patients predictively responsive to any of the three drugs ($n = 91$) than in other patients ($P = 2.00e-5$, 0.035, and $2.00e-5$, respectively) (Fig. 4b), consistent with a previous report that CIN-type gastric cancer is sensitive to adjuvant chemotherapy[34].

For targeted therapeutic agents (Fig. 4c, d), 19.4% (24/124) of our patients had *ERBB2* amplification and therefore could benefit from FDA approved *ERBB2* inhibitors, including Trastuzumab, Pertuzumab, Neratinib, Lapatinib, and Ado-Trastuzumab Emtansine. Of the 100 patients without *ERBB2* amplifications, 89 had other genomic alterations likely druggable by 59 classes of agents. Specifically, we predicted that WEE1, CDK4/6, and PARP inhibitors would be effective for 70, 42, and 27 of our patients, respectively, and together they could cover 93.3% (83/89) of our patients without *ERBB2* amplification. Among potentially actionable gene alterations found in all 124 patients (Fig. 4c), the *TP53* mutations in 88 patients were the major contributors to the predicted efficacy of WEE1 inhibitors. While only 19 patients had *CDK4/CDK6* alterations, other alterations such as *CDKN2A/CDKN2B* deletions or *CCND1* amplifications in another 29 patients might also render them vulnerable to CDK4/6 inhibitors. Using the RNA sequencing data of paired ACGEJ and adjacent normal tissue samples of each patient, we found that 85.2% (404/474) of the predictively targetable gene alterations had corresponding expression changes (Supplementary Data 6). Recurrent inconsistency (>10 samples) was only found for *FGF3* and *FGF4* amplifications (in 13 and 14 samples, respectively), potentially affecting the predicted response rates of two FGFR inhibitors Lucitanib and Dovitinib.

We compared the classes of targeted therapeutic agents with ≥20% predicted response rates in our 89 patients and 85 TCGA

ACGEJ patients that had no *ERBB2* amplifications (Fig. 4d) and found that of the eight guideline or clinical-trial drug classes (i.e., WEE1, CDK4/6, PARP, FGFR, PI3K, AKT, AURKA-VEGF, and MTOR inhibitors), WEE1 and FGFR inhibitors had higher whereas others had lower response rates in our patients than in TCGA patients. Since *CCNE1* was recurrently amplified, overexpressed, and often co-amplified with *ERBB2* in our tumor samples, we took a special interest in druggable CDK2 whose activity is regulated by Cyclin E1 produced by *CCNE1*. The predicted response rate to existing CDK2 inhibitors was 29.8% (37/124) in all our patients and 24.7% (22/89) in our patients without *ERBB2* amplifications, compared with 16.2% (17/105) and 15.3% (13/85) in TCGA ACGEJ patients, respectively.

Unfortunately, 11 of our patients had no potentially druggable gene alterations. The tumor genomes of these patients had significantly lower TMB (median 1.87 versus 0.06 per Mb, $P = 1.81e-4$), fewer CNVs at the chromosomal arm level (median 1 versus 17, $P = 7.84e-6$) and the gene level (median 0 versus 490, $P = 6.10e-6$) and less frequent WGD (2/11 versus 72/113, $P = 0.007$) than the tumor genomes of other patients (Fig. 4e). Based on the hematoxylin and eosin (H&E)-stained histopathological images of these 11 patients' tumor samples (Supplementary Fig. 1b), we estimated that the range of their tumor cell contents were 60−95%, which eliminated the possibility of normal tissue contamination.

We performed in vitro experiments to assess the vulnerabilities of cell lines with predicted druggable gene alterations to corresponding therapeutic agents. We tested six chemotherapeutic agents and five targeted therapeutic agents on eight cell lines with different predictive vulnerabilities (Supplementary Fig. 3; Supplementary Data 7) and the results were in line with the predictions (Fig. 4f). For chemotherapeutic agents, cells with oncogenic mutation in *NF1* were significantly more sensitive to Vinblastine and cells with oncogenic mutations in *ATM* and *ERCC4* were slightly more sensitive to Cisplatin than those without corresponding mutations. For targeted therapeutic agents, cells with *CCNE1* amplifications were highly responsive to CDK2 inhibitor Roniciclib. Cells with oncogenic mutations in DNA repair genes *BRCA2*, *ATM*, *ATR*, *CHEK2*, and *FANCA* were more sensitive to PARP inhibitor Olaparib and cells with oncogenic mutations in *TP53* or *BRCA1* were more sensitive to WEE1 inhibitor MK-1775 than those without corresponding mutations. Cells with *CDK4/6* or *FGF3/4* amplifications showed only slightly but significantly higher sensitivity to CDK4/6 inhibitor LEE011 or FGFR inhibitor Dovitinib, respectively, than those without corresponding gene amplifications (Fig. 4f).

**Featured genomic and transcriptomic changes are correlated with survival in patients.** Since tumor genomic and transcriptomic alterations may determine the clinical outcomes of patients, we analyzed the connections between key genomic and transcriptomic changes of ACGEJ and survival time in patients (see Methods). Regarding genomic changes, both TMB and gene level CNVs were significantly correlated with survival time: patients with low TMB (< 3.45 per Mb) or more gene level CNVs (≥382) in ACGEJ had short survival time (both log-rank $P = 0.030$) compared with patients with high TMB (≥3.45 per Mb) or less gene level CNVs (<382) in ACGEJ, and the hazard ratios (HRs) of low TMB and more gene level CNVs for death adjusted for age, sex, and tumor stage were 7.70 (95% confidence interval (CI) = 1.02–58.26) and 2.79 (95% CI = 1.05–7.42), respectively (Fig. 5a, b). Given that the TMB and gene level CNVs were moderately correlated in our patients (Spearman's $\rho = 0.32$, $P = 2.50e-4$), we further classified patients into two groups, the low-TMB high-CNV group ($n = 31$) versus others ($n = 52$), and the

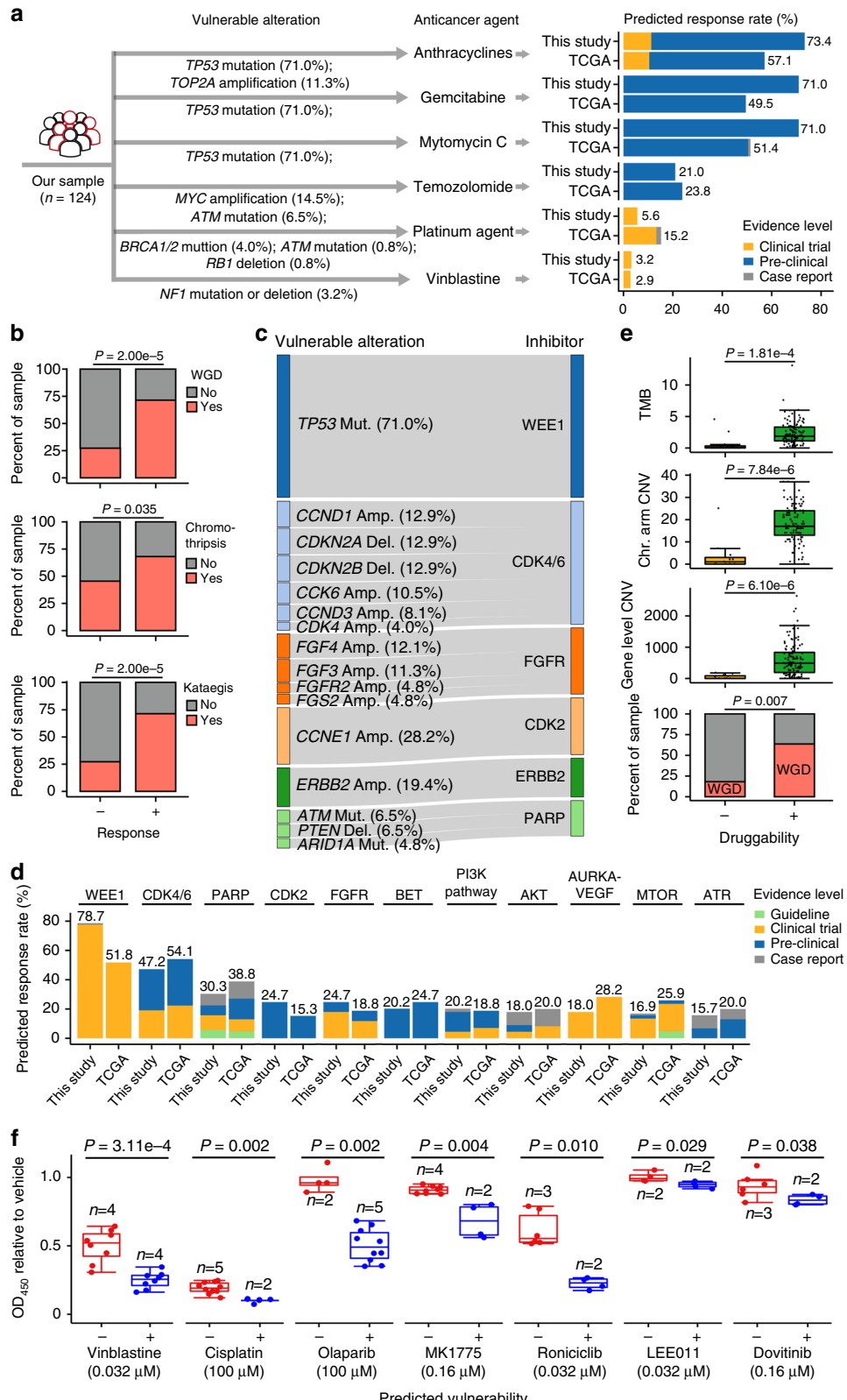

first group showed significantly shorter survival time than the second group (log-rank $P = 0.001$), with an adjusted HR of 4.61 (95% CI = 1.80–11.80) (Fig. 5c). Incorporating TMB with gene level CNVs led to a bigger HR than considering the latter alone, suggesting the two genomic alterations may combinedly affect ACGEJ survival time. We also found a significant correlation between Signature 17 activities (≥17.1% or <17.1%) in ACGEJ

and survival time in patients (log-rank $P = 0.022$; Fig. 5d). The HR of high Signature 17 activity for ACGEJ death was 3.94 (95% CI = 1.38–11.28) adjusted for age, sex, tumor stage, TMB, and gene level CNVs, indicating Signature 17 activity is an independent prognostic factor. Figure 5e shows a multivariate Cox model comparing the relative impacts of featured genomic alterations and other non-genomic factors such as for age, sex, and tumor

**Fig. 4 Genomic alterations vulnerable to existing treatment options in Chinese ACGEJ in comparison with TCGA ACGEJ. a** Chemotherapeutic agents predicted to be responsive in Chinese ACGEJ. Left panel shows genomic alterations vulnerable to each chemotherapeutic agent and right panel shows the percentage of samples responsive to each chemotherapeutic agent, with comparisons between our ACGEJ patient set and the TCGA ACGEJ dataset. **b** Bar plots comparing ACGEJ samples likely responsive and unresponsive to Anthracyclines, Gemcitabine, or Mytomycin C ($n = 91$ and 33, respectively) for the frequencies of WGD, chromothripsis, and kataegis (Fisher's exact tests). **c** Genomic alterations in our ACGEJ samples and the corresponding targeted therapeutic agents. Only genomic alterations detected in ≥5 samples and agents having potential targets in ≥20% samples are shown. **d** Percentage of samples predicted to be responsive to different targeted therapeutic agents, with comparisons between our patients and TCGA patients without *ERBB2* amplification ($n = 89$ and 85, respectively). Only agents with potential targets in ≥20% of our patients or TCGA patients are shown. **e** Comparison between likely druggable and undruggable ACGEJ samples ($n = 113$ and 11, respectively) for TMB, chromosomal (Chr.) arm or gene level CNVs (two-sided Wilcoxon rank-sum tests), and the frequencies of WGD (Fisher's exact test). **f** In vitro validation of predicted vulnerabilities to chemotherapeutic and targeted therapeutic agents. Box plots compare relative viability (expressed as $OD_{450}$ value) of different cell lines, likely vulnerable (blue) or invulnerable (red) to specified therapeutic agents, after treatment with optimal dose (in parentheses) of corresponding agents for 72 h. Each dot represents the result of one independent experiment and each experiment had three replicates on one cell line; *n* represents the number of cell lines; *P* values derived from two-sided Wilcoxon rank-sum tests. Box plots in (**e, f**) show the median (central line), the 25–75% interquartile range (IQR) (box limits), the ±1.5 times IQR (Tukey whiskers), and all data points, among which the lowest and the highest points indicate minimal and maximal values, respectively.

stage. We next evaluated the survival associations of ACGEJ transcriptomic changes, specifically, aberrant activities of cancer hallmark pathways or gene signatures quantified by GSVA scores. Patients with low GSVA scores of the IFN-α response pathway ($< -0.2256$) showed significantly shorter survival time (log-rank $P = 0.004$; Fig. 5f) than those with high scores ($\geq -0.2256$). The HR of high IFN-α response for ACGEJ death was 0.25 (95% CI = 0.09–0.66) adjusted for age, sex, and tumor stage. Because down-regulated IFN-α response pathway was significantly correlated with high Signature 17 activities (Fig. 2e), we then focused analysis on this pathway and further identified two relevant genes, *Interferon Induced Protein 44* (*IFI44*) and *IFI30 Lysosomal Thiol Reductase* (*IFI30*), whose high expressions (≥3.22 or ≥28.13 Transcripts Per Million, TPM) were significantly correlated with long survival time (log-rank $P = 6.57e-5$ and 0.043, respectively; Fig. 5g, h), with respective adjusted HRs of 0.23 (95% CI = 0.09–0.60) and 0.28 (95% CI = 0.09–0.84).

We then investigated the survival associations of these genomic/transcriptomic alterations in TCGA patients with ACGEJ or non-ACGEJ CIN-type gastric cancer ($n = 184$) in the same way (see Methods). The results showed that TCGA patients with low TMB (<1.73 per Mb) or high Signature 17 activities (≥48.5%) had significantly shorter survival time (log-rank $P = 0.039$ and 0.034, respectively) than those with high TMB (≥1.73 per Mb) or low Signature 17 activities (<48.5%); the HRs of low TMB and high Signature 17 activity being 1.78 (95% CI = 0.98–3.22) and 2.61 (95% CI = 1.26–5.38), respectively (Supplementary Fig. 4a, b). Patients with high IFN-α response ($\geq -0.4892$) or high *IFI30* expression (≥10.80 TPM) in tumors had significantly long survival time (log-rank $P = 0.019$ and 0.020, respectively), with the HRs of 0.46 (95% CI = 0.27–0.79) and 0.51 (95% CI = 0.28–0.93), respectively (Supplementary Fig. 4c, d). These results were consistent with the findings in our patient set. However, in TCGA patients, few gene level CNVs (<42) and high *IFI44* expression (≥12.09 TPM) were associated with short survival time (log-rank $P = 0.026$ and 0.023), with the HRs of 2.45 (95% CI = 1.21–4.97) and 1.76 (95% CI = 1.07–2.88), respectively (Supplementary Fig. 4e, f), which were inconsistent with the results in our patients. We also assessed the survival associations of *IFI44* and *IFI30* expressions using the gene expression data of gastric cancer patients registered in the Gene Expression Omnibus (GEO). Low *IFI44* expression was consistently and significantly correlated with short survival time (log-rank $P = 0.004$, HR = 0.75, 95% CI = 0.61–0.91; Supplementary Fig. 4g), while low *IFI30* expression was marginally correlated with short survival time (log-rank $P = 0.055$, HR = 0.84, 95% CI = 0.71–1.00; Supplementary Fig. 4h).

## Discussion

ACGEJ is on the rise worldwide, but its molecular characteristics have never been independently profiled and reported, which is impeding clinical treatment and drug development of this malignancy. In the present study, we have assembled and comprehensively analyzed the genome and transcriptome data of totally 124 ACGEJ samples exclusively collected from Chinese patients. The results have indicated that, like Caucasian patients, the ACGEJ genomes of our patients were dominated by CIN-promoted tumorigenic focal CNVs while lacking recurrently mutated coding genes other than *TP53*. Both TMB and gene level CNVs of the ACGEJ genome indicate the prognosis of Chinese patients. Compared with previous reports, the present study has two major advances. Firstly, our integrative analyses on genome and bulk-tissue transcriptome sequencing data have led us to more reliable findings. Secondly, we have gone beyond descriptive genomic profiling to predict effective therapeutic agents and prognosis of patients based on their genomic and transcriptomic changes. The comprehensive findings in the present study may improve our current knowledge of ACGEJ and have implications for clinical care of patients.

In the ACGEJ genomes of our patients, *TP53* is the only significantly and recurrently mutated coding gene we have detected. That said, there may be mutated non-coding drivers, such as CTCF-binding-site SNVs attributable to COSMIC Signature 17. As one of the hallmark signatures of upper-gastrointestinal adenocarcinoma, Signature 17 is believed to arise from misincorporation of oxidized DNA precursors into genomic DNA[12,35]. It has been shown that oxidative stress may generate more mutations in introns and intergenic regions than in exons and promoters[36], and improperly handled oxidative DNA damages can lead to CIN phenotypes in model systems[37,38] and in human cells[39]. Our results are in line with these previous reports, indicating that Signature 17 SNVs may be attributed to oxidative stress and represents an etiological factor forming CIN in ACGEJ genomes. In addition, we have notably provided bulk transcriptomic evidence to support the role of Signature 17 by correlating its activities with a gene expression signature of CIN. More importantly, our analysis on bulk RNA sequencing data has revealed an immunosuppressive microenvironment in tumors with high activities of Signature 17. These results indicate that Signature 17 related genome and transcriptome changes play very important roles in the development and progression of ACGEJ, which may be of clinical relevance.

Comparing the genomic and transcriptomic alterations in our samples with those in TCGA ACGEJ samples (mostly from Caucasian patients) revealed similar CNV patterns and pathway aberrations. However, tumor samples of our patients seem to

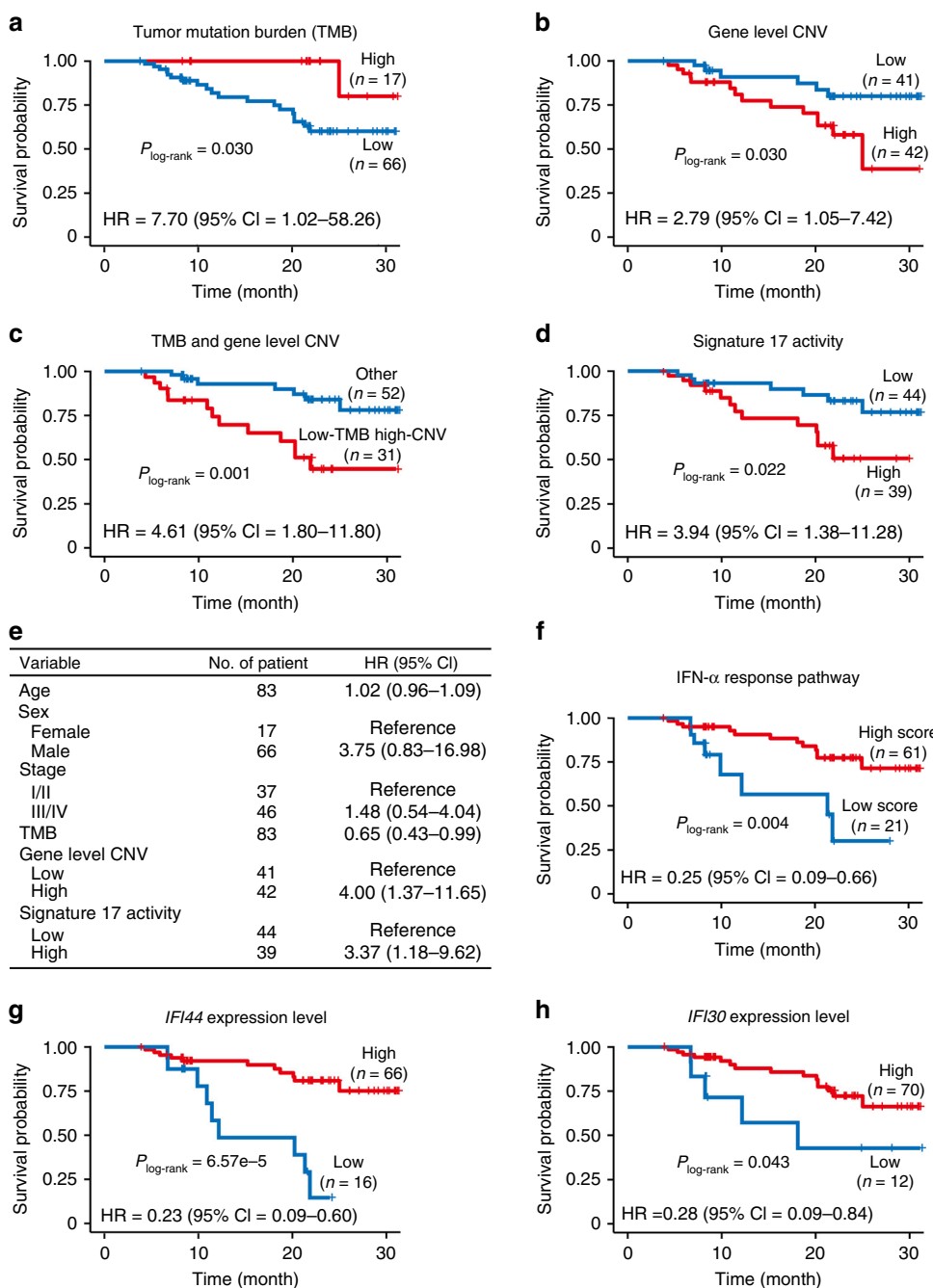

**Fig. 5 Survival-associated genomic and transcriptomic features of ACGEJ. a–d** Kaplan-Meier (KM) curves of patient survival according to TMB (**a**), gene level CNVs (**b**), TMB and gene level CNVs combined (**c**), and Signature 17 activities (**d**); *P* values derived from log-rank tests. **e** Multivariate Cox regression analysis of the effects of different genomic alterations on ACGEJ patient survival. **f**–**h** KM curves of patient survival according to the GSVA scores of IFN-α response pathway (**f**) and the expression levels of *IFI44* (**g**) or *IFI30* (**h**) gene; *P* values derived from log-rank tests. Also present with each KM plot are the hazard ratio (HR) and 95% confidence interval (CI) from multivariate Cox proportional hazard models including age, sex, tumor stage, TMB, and gene level CNVs as covariates wherever necessary.

have fewer functional mutations than TCGA samples, which might reflect different genetic and environmental backgrounds between Chinese patients and Caucasian patients or might be simply due to different coverage of our WGS data and TCGA whole-exome sequencing data. We could reduce the confounding effect of different sequencing coverage by down-sampling the reads, but that requires access to controlled TCGA data. There are also some differences in the alteration rates of candidate driver genes (e.g., *LIPF*), potential drug targets (e.g., *CCNE1*), and survival-associated immune pathways (e.g., IFN-α response)

across two cohorts. In these regards, the current study has increased general knowledge on the genetic and molecular basis of ACGEJ and, particularly, filled in the gap for non-Caucasian patients. Thus, our findings could contribute to developing more specific and effective precise care of ACGEJ worldwide.

Because there have been no specific and effective drugs for ACGEJ treatments, we have screened the ACGEJ genome and transcriptome of each patient for changes that may indicate the patient's responses to therapeutic agents currently used for other types of cancer or under clinical trials. Since DNA repair

deficiency caused by *TP53* inactivation and other events is the major mechanism associated with the efficacy of cytotoxic anticancer drugs, we have predicted that more frequent *TP53* inactivating mutations may lead to generally higher response rates to cytotoxic anticancer agents in Chinese patients than in Caucasian patients, suggesting that these chemotherapeutic agents may generally benefit Chinese patients with ACGEJ. We have also found that a considerable proportion of Chinese ACGEJ patients might benefit from targeted anticancer agents[40] such as ERBB2, WEE1, and CDK4/6 inhibitors. The predicted response rates to these agents are somewhat different between Chinese and Caucasian ACGEJ patients. Specifically, we have shown that *CCNE1* amplification is a CIN-associated oncogenic event prevalent in Chinese patients and often coexisting with *ERBB2* amplifications. Given that *CCNE1* amplification has been linked to the resistance to *ERBB2* inhibitors[41,42], it is worthwhile to conduct clinical trials for CDK2 inhibitors[43,44] in Chinese ACGEJ patients. We have validated the reliability of gene-alteration-based drug vulnerability prediction by in vitro assays in a panel of cancer cell lines. Thus, our predictions on the response rates to currently used cytotoxic anticancer agents or targeted drugs approved or tested for other types of cancer have provided molecular evidence for their clinical use or clinical trials for treating ACGEJ.

In the present study, we have identified several prognostic markers from genomic or transcriptomic changes associated with the survival time in ACGEJ patients. At the genomic level, we have found that TMB and gene level CNVs are independently and combinedly correlated with the survival time in our patients. In addition, we have identified Signature 17 activity as another independent marker for ACGEJ survival time, which one may expect because in our patient set Signature 17 is correlated with CIN-related genomic changes such as WGD and chromothripsis, both confer terrible malignant cancer phenotypes[23,45]. Consistent with our result, a previous genomic study in breast cancer has also shown that Signature 17 is enriched in metastatic tumors and linked to poor prognosis of patients[46]. Nevertheless, in the present study we went beyond genome data and incorporated transcriptome data for analysis. As a result, we have found that a Signature 17-correlated cancer hallmark pathway, IFN-α response, is also significantly associated with the survival time in our ACGEJ patient set. Moreover, transcriptome analysis has shown that tumors with high Signature 17 activities had repressed activities of IFN-producing cytotoxic cells. These findings emphasize that immunosuppression in the tumor microenvironment, likely induced by high Signature 17 activities in cancer cells, plays an important role in the progression of ACGEJ. Specifically, we have found significant associations of high expression levels of *IFI44* and *IFI30*, two IFN-α-inducible genes, with good survival in our patient set. Our comparative analyses on TCGA and GEO datasets have yielded both consistent and inconsistent results. The associations of TMB levels, Signature 17 activities, IFN-α response activities, and *IFI30* expression with patient survival time are quite consistent in our patient set and TCGA and GEO patient set. The results for gene level CNVs in our cohort and the TCGA cohort seem contradictory. For *IFI44*, the GEO data agreed with our finding, whereas the TCGA data showed a significantly negative correlation between high *IFI44* expression and patient survival. *IFI44* has been reported to function in many biological processes through IFN signaling pathways, including anti-proliferative activity in tumors[47,48]. High *IFI30* expression in tumor cells has been linked to improved cancer survival[49,50], possibly due to its ability to enhance the presentation of tumor antigens for T cell recognition[51] or to regulate the cellular redox state and proliferation[52]. Thus, the identified down-regulation of *IFI44* and *IFI30* might be a simplified prognostic marker for ACGEJ in clinical use.

Despite the advances discussed above, we acknowledge some limitations in the present study. Firstly, we focused on genomic alterations in coding sequences and barely touched upon noncoding elements, which may harbor additional ACGEJ drivers. Secondly, we have not fully investigated the extent of intratumor heterogeneity and how it may affect our findings. Both issues are on our agenda for future studies. In addition, the clinical or preclinical evidence for the anticancer agents we referred to mostly came from studies on Caucasian patients and thus it is uncertain whether and how much they will apply to Chinese patients.

In conclusion, our present study has profiled ACGEJ in Chinese patients as a CNV-dominant CIN-type tumor, revealed the types and the distributions of various druggable changes in tumor genomes, and identified genomic and transcriptomic prognostic markers that have potential clinical implications. These findings have furthered our understanding on ACGEJ and would help develop more effective therapeutic strategies to precisely fight this malignancy, especially for Chinese patients.

## Methods

**Biospecimen and clinical data**. The biospecimen used in this study were obtained from 124 Chinese ACGEJ patients (Supplementary Table 3) recruited at the Linzhou Cancer Hospital and Linzhou Esophageal Cancer Hospital (Henan Province, China) between 2013 and 2018. ACGEJ tumor, adjacent non-tumor gastric tissue (≥5 cm from tumor margin), and peripheral blood samples were collected at the time of surgical resection. ACGEJ (tumors arising at the gastric cardia and/or gastroesophageal junction with/without involvement of other esophageal and/or gastric subsites) was confirmed by two pathologists via histopathological examination. No patient had received chemotherapy or radiotherapy before surgery. Clinical data were collected from the medical record of each patient. We conducted follow-up phone interviews with 83 patients in a period of 20.2 months (median). At the most recent interview (November 2018), 75.9% (63/83) of patients were still alive. This study was approved by the Institutional Review Board of Cancer Hospital, Chinese Academy of Medical Sciences. Written informed consent was solicited from every patient prior to sample collection.

**DNA and RNA sequencing**. Only tumor samples containing ≥ 60% of cancer cells evaluated by examining tissue slides (Supplementary Fig. 1a) were used for DNA and RNA sequencing. WGS data were generated from matched tumor and blood samples from 124 patients. The bulk RNA sequencing data were generated from matched tumor and normal tissue samples from 123 of the 124 patients. DNA and total RNA were extracted from tissue samples using the AllPrep DNA/RNA Kit (Qiagen). Blood DNA was extracted using QiaAmp Blood Midi Kit (Qiagen). Library preparation was as previously described[53]. All libraries were sequenced on Illumina HiSeq xTen in 2 × 150 bp paired-end mode.

**Detection of somatic SNVs, indels, CNVs, and SVs**. DNA sequence reads were aligned to the Ensembl GRCh37 human genome using BWA-MEM (v0.1.22)[54]. Somatic mutations (SNVs and indels) were detected using Strelka2 (v2.8.3)[55] and annotated by Ensembl Variant Effect Predictor (VEP, release 90)[56]. We filtered the results with gatk-tools (v0.2.2). TMB was measured by the number of non-silent SNVs/indels. Somatic CNVs (based on the log ratio of tumor to non-tumor reads) were detected from WGS data by BIC-seq2 (v0.2.4)[57]. Allelic copy number, ploidy, and purity were estimated by FACETS (v0.5.14)[58]. Large-scale (>50 bps) SVs were independently called from the WGS data by Delly (v0.7.3)[59], GRIDSS (v2.4.0)[60], Manta (v1.1.1)[61], and svABA (v0.2.1)[62]; then we used SURVIVOR (v1.0.6)[63] to merge nearby break points and filter out SVs detected by only one tool.

**Identification of specific genomic features**. We used MutSigCV (v1.41)[64] to identify genes with significantly recurrent coding-sequence SNVs/indels and ActiveDriverWGS (v1.0.1)[65] to identify non-coding genomic elements (e.g., promoters, non-coding RNAs) with significantly more unexpected somatic SNVs. These genes and genomic elements were deemed potential drivers of ACGEJ. Functional mutations were defined as SNVs/indels likely affecting the function of a gene, including in-frame and frame-shift indels, nonsense SNVs, OncoKB annotated oncogenic or likely oncogenic missense SNVs, or SNVs/indels occurring on splice site, start codon, or stop codon.

Segmentation files output by BIC-seq2 (for our samples) or downloaded from the TCGA website (excluding the regions within 3 Mb around centromeres and 1 Mb at both chromosome ends) were used as input for GSITIC2.0 (v.2.0.23)[66] to quantify the CNV status of each gene in each tumor genome. In a tumor genome, we considered a gene undergoing homozygous deletion, copy-number loss, copy-number gain, or amplification if the corresponding GISTIC score = −2, ≤ −1, ≥ 1, or = 2, respectively. Significantly recurrent focal CNV regions (Supplementary Data 8, 9) were also determined by GISTIC2.0 (FDR $q < 0.25$).

We considered a chromosomal arm undergoing copy-number gain or loss if ≥2/3 genes on that arm had GISTIC score ≥1 or ≤ −1, respectively. We defined arm level CNVs as chromosomal arms with copy-number gain or loss and gene level CNVs as gene amplifications or deletions. Only genes with GISTIC scores ±2 were counted when measuring gene level CNVs. We considered an ACGEJ genome undergoing WGD if ≥50% autosomes had a major allele copy number ≥2; the 50% threshold was determined following a previous study[23] (Supplementary Fig. 5). We measured the proportion of CNV affected genome using regions where | log2(reads ratio) | ≥0.3. Kataegis events were detected following the method of a previous WGS study[67]. Chromothripsis events were identified by ShatterSeek (v0.4)[45] based on the merged SV results and FACETS CNV results.

**Identification of mutational signatures**. We started by extracting de novo signatures from and inferring the prevalence of 30 published COSMIC mutational signatures in the Chinese cohort. Both had led us to recognize the prevalent activities of COSMIC Signatures 1, 3, 5, 8, and 17 (or their analogues). Thus, we only focused on these five signatures and clamped the activities of other signatures to zero in all subsequent analyses. We used deconstructSigs (v1.8.0)[68] to measure the activity of every signature in every tumor genome (the percentage of SNVs across the exome or genome attributable to the signature). For each somatic SNV, we determined its clonality and the mutational signature it most likely belongs to using Palimpsest (v.2.0.0)[69].

**Analysis of bulk-tissue RNA sequencing data**. We used HISAT2 (v2.1.0)[70] to align RNA sequence reads to the Ensembl GRCh37 human genome and then used StringTie (v1.3.3b)[71] to re-assemble transcriptome and quantify the expression level of each gene in each sample. The assembly process was guided by the reference annotations from GENCODE (v19)[72]. Gene expression levels were quantified by Transcript per Million (TPM). We considered a gene differentially expressed in tumor and normal samples if the log2 + 1 transformed TPM change was significant in a paired Student's $t$-test (FDR $q < 0.05$) and the fold change of mean TPM was >1.2 or <0.8. In other comparing situations, we considered a gene differentially expressed in two groups of samples if the TPM change was significant in a paired Wilcoxon test ($P < 0.05$) and the fold change of median TPM was >1.2 or <0.8. The activity of specific pathway or gene signature in each tumor or normal sample was quantified using the R package GSVA (v1.30.0)[31] on log2 + 1 transformed TPM. The differential expression of specific pathway or gene signature between tumor and normal samples was assessed using gene set enrichment analysis (GSEA)[73] (implemented in the R package clusterProfiler v3.10.1[74]). We quantified the fractions of 10 immune cell types (i.e., macrophages (M1 and M2), monocytes, neutrophils, B, NK, CD8 + /CD4 + /regulatory T, myeloid dendritic cells, and uncharacterized cells) in individual tumor and adjacent normal samples using quanTIseq[75] (implemented in the R package immunedeconv v2.0.0).

**Identification of Signature17-correlated pathways and gene signatures**. We identified these pathways and gene signatures using matched WGS and bulk RNA-seq data generated from 123 patients. We first calculated partial Spearman's correlation (using the R package ppcor v1.1) between Signature 17 activities and the expressions of each gene across tumor samples, controlling for tumor purity and the activities of other four signatures (i.e., Signatures 1, 3, 5, and 8). Then we examined known cancer hallmark pathways and gene signatures obtained from a previous study[30] one by one to see whether they are enriched with Signature 17-correlated genes using GSEA on the calculated correlation coefficients.

**Analysis and integration of public data**. To fairly compare the transcriptomic profiles of our ACGEJ samples and TCGA samples, we combined the gene expression data (measured by TPM) of both cohorts and that of normal gastroesophageal junction tissue samples from the Genotype-Tissue Expression (GTEx) Project, performed normalization and batch correction and used the processed data for comparative analyses. We normalized the combined gene expression dataset by keeping genes whose expression levels were quantified in all three individual datasets ($n = 19{,}587$) and rescaling the TPM values per sample to maintain a sum of 1 million. The ComBat function implemented in the R package sva (v3.28.0)[76] was used to remove the batch effect of different data sources (i.e., data source as a batch parameter). Tumor and normal samples in the combined dataset were respectively grouped together and appropriately separated from each other (Supplementary Fig. 6). The GSVA scores of featured pathways and gene signatures in either cohort before and after the processing were highly correlated; so were the estimated fractions of different immune cell types.

**In vitro validation of drug vulnerability prediction**. We used eight human cancer cell lines including the ACGEJ cell line OE19 (purchased from Beijing Beina Chuanglian Biotechnology Institute), esophageal adenocarcinoma cell lines OE33 and SK-GT-4 (purchased from Nanjing COBIOER Biosciences Company Limited), gastric adenocarcinoma cell lines AGS and HGC-27, and colorectal cancer cell lines HCT-116, LoVo and RKO (purchased from the Cell Bank of Type Culture Collection of Chinese Academy of Sciences Shanghai Institute of Biochemistry and Cell Biology). We obtained somatic mutation and CNV data of these cell lines from COSMIC Cell Lines Project (https://cancer.sanger.ac.uk/cell_lines) and predicted

each cell line as vulnerable or invulnerable to specific chemotherapeutic or targeted therapeutic agents based on whether it carried corresponding gene alterations. The efficacy of 11 drugs was respectively assessed by cell viability measured using the CCK-8 kit (Dojindo Labs). Briefly, cells were treated with optimal dose of each drug and the viability was measured after incubation for 72 h. All analyses were performed in two independent experiments and each had three replicates.

**Survival analysis**. We used the log-rank test in univariate survival analyses and the Cox proportional hazards model in multivariate survival analyses (both implemented in the R package survival v2.43-3). The Kaplan-Meier plot was used for presentation. The specific cutoff we used to dichotomize a continuous variable (e.g., gene level CNVs, Signature 17 activities, IFI44 expression) and then group patients was determined by testing a series of values of that variable with fixed increments and then choosing the one by which both log-rank and Cox $P$ were <0.05 and the log-rank $P$ was minimized. This process would not give us any options if there was no statistically significant result in the first place. The cutoffs we used to obtain the presented results were 3.45 per Mb for TMB, 382 for gene level CNVs, 17.1% for Signature 17 activities, −0.2256 for IFN-α response, 3.22 for *IFI44* expressions, and 28.13 for *IFI30* expressions. The same approach was applied to divide TCGA patients into good or poor survival groups, with the specific cutoffs of 1.73 per Mb for TMB, 42 for gene level CNVs, 48.5% for Signature 17 activity, −0.4892 for IFN-α response, 12.09 for *IFI44* expressions, and 10.80 for *IFI33* expressions.

**Identification of prognostic marker genes**. To identify prognostic marker genes in the IFN-α response pathway, we first selected 40 genes whose expressions were (a) significantly correlated with the pathway GSVA scores across tumor samples (| Spearman's $\rho$| > 0.3, $P < 0.05$) and (b) significantly different between the two survival groups divided based on their IFN-α pathway GSVA scores ($t$-test $P < 0.05$). Next, we tested each of the 40 genes and found 13 of them were significantly correlated with patient survival (with at least one cutoff under which log-rank and Cox $P < 0.05$). We finally picked *IFI44* and *IFI30* out of the 13 genes using a Lasso-Cox method as implemented in the glmnet package v3.0-2[77].

**Statistical analysis**. We used Fisher's exact test for any independence test between two categorical variables and Wilcoxon rank-sum test for any independence test between a continuous variable and a binary categorical variable, when there was no covariate to adjust for. Otherwise, we used an $F$-test to compare two generalized linear models, one of which included the variable being evaluated as a predictor. Spearman's rank correlation coefficient was used to measure the correlation between two continuous variables.

**Reporting summary**. Further information on research design is available in the Nature Research Reporting Summary linked to this article.

## Data availability
The raw WGS and RNA sequencing data generated in this study are deposited in the Genome Sequence Archive of Beijing Institute of Genomics, Chinese Academy of Sciences (http://bigd.big.ac.cn/gsa/, accession number HRA000025). The gene expression data are also publicly available from NCBI Gene Expression Omnibus (https://www.ncbi.nlm.nih.gov/geo, accession number GSE159721). The whole-genome somatic variants are also publicly available from the European Variation Archive (https://www.ebi.ac.uk/eva, accession number PRJEB41070). We obtained somatic mutation and CNV data of TCGA ACGEJ samples[8] from the Broad Institute GDAC Firehose website (https://gdac.broadinstitute.org/) and somatic mutation data of additional 46 ACGEJ samples[15,16] from the Tumor Portal (http://www.tumorportal.org). The gene expression (TPM) data of TCGA ACGEJ samples[10] were obtained from TCGA Pan-cancer Atlas publication web page (https://gdc.cancer.gov/about-data/publications/pancanatlas). The gene expression (TPM) data of normal gastroesophageal junction tissue samples[78] were obtained from the GTEx Portal (http://www.gtexportal.org/home/). Additional survival analyses in the GEO datasets were conducted on the Kaplan-Meier Plotter website (https://kmplot.com)[79] with automatically selected best cutoffs. Other data that support the findings of this study are available within the supplementary files or available from the authors upon request.

## Code availability
We used published software for all our analyses as indicated. Other accompanying code is available from the authors upon request.

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

## Acknowledgements

Supported by National Key Research and Development Program of China (2016YFC1302700 to C.W.), National Science Fund for Distinguished Young Scholars (81725015 to C.W.), Medical and Health Technology Innovation Project of Chinese Academy of Medical Sciences (2016-I2M-3-019 to D.L.; 2016-I2M-4-002 to C.W.; 2019-I2M-2-001 to D.L. and C.W.; 2016-I2M-1-001 to W.T.), Beijing Outstanding Young Scientist Program (BJJWZYJH01201910023027 to C.W.) and National Natural Science Foundation of China (81988101 to D.L. and C.W.).

## Author contributions

C.W., G.G., and D.L. conceptualized and supervised this study. Y. Lin, Y.S., and Y.M. performed bioinformatics analysis. Y. Lin, Y. Luo, and W.G. performed statistical analysis. X.Z. performed in vitro drug vulnerability experiments. Y. Luo and Y.X. prepared DNA and RNA samples. Y.X., M.S., and W.T. responded for clinical data and biospecimen collection and performed experiments. Y. Lin and Y. Luo drafted and C.W., G.G., and D.L. reviewed the manuscript.

## Competing interests

The authors declare no competing interests.
