## [Peer Review File · Nature Communications]

REVIEWER COMMENTS

Reviewer #1 (Remarks to the Author): Expert in GI cancer

Intestinal type of esophageal cancer occurs at the GE junction and has poor survival rate. It is critical to discover the molecular and genomic markers that can help improve the accuracy of early detection and precision of tailored therapeutics. Lin et al employed the bioinformatics methods to study a cohort of Chinese patients with esophageal adenocarcinoma and confirmed previous work on CIN-associated focal CNVs and Signature 17. Furthermore, the authors claimed that they identified genomic and transcriptomic prognostic markers of potential clinical values.

The major weakness of Liu et al is the lack of employment of any biological system in this study. Thus, the correlation between genomic alterations and the sensitivity to therapeutic agents is interesting but remains theoretical. There is no patient-derived cell lines being generated and used to test the prediction. In addition, for the patients included in this study, what treatments did they have and how did they respond?

The RNAseq was done on tumor and matched normal region that is 5cm away from the tumor. However it is unclear whether the authors refer to normal esophageal or gastric tissues.

For the 11 of patients that have significantly lower TMB, and fewer CNV, the authors did not comment whether it is possibly due to the contamination of normal tissues in the samples. It would be in general useful if the authors can show the histology of the tumors that they studied, at least some examples.

Reviewer #2 (Remarks to the Author): Expert in genomics

In this study, Lin et al., presented genomic and transcriptomic alterations in ACGEJ from large number of Chinese patients. They further analyzed their drug vulnerabilities and associations with the survival time. They highlighted the major genomic changes of Chinese ACGEJ patients are chromosome instability promoted tumorigenic focal copy-number variations and COSMIC Signature 17-featured single nucleotide variations. Most interestingly, they demonstrated the potential drug vulnerabilities in this patient cohort and TCGA samples. The study is overall interesting, and represent a significant resource to further develop effective therapeutic strategies in ACGEK patients.

Major comments:

1. While it is nice to show the mutation landscape/CNV/transcriptomic alterations in the current patient cohort, it will be interesting to investigate the similarity/difference between Chinese patients and TCGA patients.
2. The section of potential drug vulnerabilities is particular interesting. An experimental validation of one or two selected drug in related cancer cell lines (if applicable) will be great.
3. Further in this section, the authors showed quite different of response between the current

patient cohort and TCGA – can this be explained by the difference of the mutation/CNV/transcriptomic alterations?

4. For the survival analysis, the authors divided samples into two groups. Could the authors specify the reason for those cutoffs, for example, 3.45 per MB for TMB, 382 for gene level CNVs, 17.1% of signature 17 activities, and -0.2256 for IFN- α response, etc.

5. For those genomic and transcriptomic changes associated with survival in this study, are these alterations also associated with survival in TCGA cohort?

Minor comments:

For those drugs that may have a higher response rate in Chinese patients, could the authors discuss the future potential for these drugs in clinical practice?

Responses to the Reviewers' Comments

Reviewer #1

General comment

Intestinal type of esophageal cancer occurs at the GE junction and has poor survival rate. It is critical to discover the molecular and genomic markers that can help improve the accuracy of early detection and precision of tailored therapeutics. Lin et al employed the bioinformatics methods to study a cohort of Chinese patients with esophageal adenocarcinoma and confirmed previous work on CIN-associated focal CNVs and Signature 17. Furthermore, the authors claimed that they identified genomic and transcriptomic prognostic markers of potential clinical values.

Response: We are grateful for these positive general comments.

Specific Comments

Comment 1: The major weakness of Lin et al is the lack of employment of any biological system in this study. Thus, the correlation between genomic alterations and the sensitivity to therapeutic agents is interesting but remains theoretical. There is no patient-derived cell lines being generated and used to test the prediction.

Response 1: Thank you for the comment. Per the suggestion we have performed additional experiments to address this issue. We have used human ACGEJ cell line OE19, esophageal adenocarcinoma cell lines SK-GT-4 and OE33, gastric adenocarcinoma cell lines HGC-27 and AGS and colorectal cancer cell lines RKO, LoVo and HCT-116, which had different gene alterations potentially vulnerable to chemotherapeutic agents or target therapeutic agents, to test the reliability of gene-alteration-based drug vulnerability predictions. We found that the in vitro assay results are consistent with the computational predictions. We have added the experimental validation results as the last paragraph of the 'Genomic alterations indicate potential vulnerability to existing therapeutic agents' section of RESULTS in the revised manuscript (page 12, middle; Figure 4f). Additionally, we have added a new section to METHODS named 'In vitro validation of drug vulnerability prediction' (pages 23–24) and a sentence to DISCUSSION (page 17, the second last sentence of the first paragraph). We hope these additions have addressed your concern and improved our paper.

Comment 2: In addition, for the patients included in this study, what treatments did they have and how did they respond?

Response 2: All patients included in this study had only received surgery when we collected their tumor samples. Because this study intended to investigate the genomic and transcriptomic features of untreated ACGEJ, we made sure the enrolled patients received no other anticancer treatment such as chemoradiotherapy before surgery, as stated in our original manuscript (page 19, lines 7–8 in the 'Biospecimen and clinical data' section).

Comment 3: The RNAseq was done on tumor and matched normal region that is 5cm away from the tumor. However it is unclear whether the authors refer to normal esophageal or gastric tissues.

Response 3: We are sorry for the missing information. The matched non-tumor tissues used for

RNA sequencing were adjacent normal gastric tissues 5 cm away from tumors. We have provided this information in the revised manuscript (page 19, the third line in the 'Biospecimen and clinical data' section).

Comment 4: For the 11 of patients that have significantly lower TMB, and fewer CNV, the authors did not comment whether it is possibly due to the contamination of normal tissues in the samples. It would be in general useful if the authors can show the histology of the tumors that they studied, at least some examples.

Response 4: All tumor samples used for sequencing contain $\geq 60\%$ cancer cells, estimated based on the hematoxylin and eosin (H&E)-stained images of tumor tissues. We have confirmed that the tumor samples of the 11 patients with significantly lower TMB and fewer CNVs than others had approximate tumor cell contents ranging from 60% to 95%, thus eliminating the possibility of normal tissue contamination. We have added these data to the revised manuscript (page 4, middle; page 12, the last sentence of the first paragraph; page 19, the first sentence of the 'DNA and RNA sequencing' section) and have shown some representative tumor histopathology images in Supplementary Figure 1 as per your suggestion.

Reviewer #2

General comment

In this study, Lin et al., presented genomic and transcriptomic alterations in ACGEJ from large number of Chinese patients. They further analyzed their drug vulnerabilities and associations with the survival time. They highlighted the major genomic changes of Chinese ACGEJ patients are chromosome instability promoted tumorigenic focal copy-number variations and COSMIC Signature 17-featured single nucleotide variations. Most interestingly, they demonstrated the potential drug vulnerabilities in this patient cohort and TCGA samples. The study is overall interesting, and represent a significant resource to further develop effective therapeutic strategies in ACGEJ patients.

Response: We appreciate your positive comments and are happy to address the following questions.

Major comments:

Comment 1: While it is nice to show the mutation landscape/CNV/transcriptomic alterations in the current patient cohort, it will be interesting to investigate the similarity/difference between Chinese patients and TCGA patients.

Response 1: As per the suggestion, we have compared major mutations, CNVs and transcriptomic alterations of our patient set and the TCGA ACGEJ patient set. The results show overall similarity with some differences and have been presented in a new section 'Comparative analysis on major genomic alterations of our patients and TCGA patients' of RESULTS (pages 8–10). We have also discussed the comparison results (page 16), specified the used methods (pages 22–23) and changed figure and citation numbers accordingly in the revised manuscript.

Comment 2: The section of potential drug vulnerabilities is particular interesting. An experimental validation of one or two selected drug in related cancer cell lines (if applicable) will be great.

Response 2: Per the suggestion we have performed additional experiments to address this issue. We have used human ACGEJ cell line OE19, esophageal adenocarcinoma cell lines SK-GT-4 and OE33, gastric adenocarcinoma cell lines HGC-27 and AGS and colorectal cancer cell lines RKO, LoVo and HCT-116, which had different gene alterations potentially vulnerable to chemotherapeutic agents or target therapeutic agents, to test the reliability of gene-alteration-based drug vulnerability predictions. We found that the in vitro assay results are consistent with the computational predictions. We have added the experimental validation results as the last paragraph of the 'Genomic alterations indicate potential vulnerability to existing therapeutic agents' section of RESULTS in the revised manuscript (page 12, middle; Figure 4f). Additionally, we have added a section to METHODS named 'In vitro validation of drug vulnerability prediction' (pages 23–24) and a sentence to DISCUSSION (page 17, the second last sentence of the first paragraph). We hope these additions have addressed your concern and improved our paper. Thank you.

Comment 3: Further in this section, the authors showed quite different of response between the current patient cohort and TCGA; can this be explained by the difference of the mutation/CNV/transcriptomic alterations?

Response 3: Yes. We predicted drug responses based on potentially druggable functional somatic mutations and deep CNVs of specific genes, so the predicted patient response differences indeed reflect different druggable mutations/CNVs in their tumor samples. As stated in the original manuscript (page 11, the second last sentences of the first paragraph), "we found that 85.2% (404/474) of the predictively targetable gene alterations had corresponding expression changes in our patient cohort." In the experimental validation we have performed, drug responses were predicted in the same way, i.e., based on potentially druggable mutations and deep CNVs of the cell lines we have used, as described in the added 'In vitro validation of drug vulnerability prediction' section of METHODS in the revised manuscript (pages 23–24).

Comment 4: For the survival analysis, the authors divided samples into two groups. Could the authors specify the reason for those cutoffs, for example, 3.45 per MB for TMB, 382 for gene level CNVs, 17.1% of signature 17 activities, and -0.2256 for IFN- α response, etc.

Response 4: The specific cutoff we used to dichotomize a continuous variable (e.g., TMB, gene level CNVs, Signature 17 activities, IFN- α response) and then group patients was determined by testing a series of values of that variable with fixed increments and then choosing the one by which both log-rank and Cox P were < 0.05 and the log-rank P was minimized. This process would not give us any options if no statistically significant result was available in the first place. We have extended the 'Survival analysis' section of METHODS in the revised manuscript (page 24, middle) to include the specific cutoffs we used to obtain the presented results.

Comment 5: For those genomic and transcriptomic changes associated with survival in this study, are these alterations also associated with survival in TCGA cohort?

Response 5: Per the comment, we have tested the associations of these alterations with survival in TCGA patients. There were both positive and negative results, which we have incorporated into the reconstructed 'Featured genomic and transcriptomic changes are correlated with survival in

patients' section of RESULTS (page 14, the last paragraph before DISCUSSION). In addition, we have discussed the results (page 18, top) and provided analysis details in the 'Survival Analysis' section of METHODS in the revised manuscript (page 24, middle).

Minor comments:

For those drugs that may have a higher response rate in Chinese patients, could the authors discuss the future potential for these drugs in clinical practice?

Response: In the original manuscript, we discussed and advocated clinically testing CDK2 inhibitors in Chinese patients (page 17, lines 1–5). Per the comment, we have added '...suggesting that these chemotherapeutic agents may generally benefit Chinese patients with ACGEJ' (page 16, the last paragraph) to state the future potential of using cytotoxic anticancer reagents in clinical practice in the revised manuscript. We hope this revision is satisfactory.

REVIEWERS' COMMENTS

Reviewer #1 (Remarks to the Author):

The authors have provided satisfactory responses and changes to the manuscript. The manuscript is much improved and has my full support.

Reviewer #2 (Remarks to the Author):

The authors did an excellent job to revise the manuscript, and I have no more concerns.

Responses to the Reviewers' Comments

Reviewer #1 (Remarks to the Author):

The authors have provided satisfactory responses and changes to the manuscript. The manuscript is much improved and has my full support.

Reviewer #2 (Remarks to the Author):

The authors did an excellent job to revise the manuscript, and I have no more concerns.

Response: We thank both reviewers for your help and support.